# ONE MEASURE, MANY BOUNDS: BRIDGING TV, VARIANCE, AND MUTUAL INFORMATION

## ABSTRACT

Understanding the generalization of machine learning algorithms remains a fundamental challenge. While mutual information provides a powerful lens for analysis, we introduce a more flexible, one-parameter family of information-theoretic generalization bounds based on the vector-valued $L_p$-norm correlation measure, $V_\alpha$. Our framework unifies and interpolates between several existing information-theoretic guarantees, including those based on total variation and Rényi information. The primary conceptual contribution of our work emerges at $\alpha = 2$, where our framework yields a novel and intuitive variance-based bound. This result establishes the variance of the algorithm's output distribution, $\mathrm{Var}_S[p(w|S)]$, as a direct, data-dependent measure of algorithmic stability. We prove that this measure directly controls the generalization error, thus providing a new, information-theoretic perspective on how unstable (high-variance) algorithms fail to generalize. Extensive simulations demonstrate that our bounds, particularly for $\alpha = 2$, can be significantly tighter than classical mutual information guarantees.

## 1 INTRODUCTION

Understanding why large, over-parameterized models generalize well despite their capacity to memorize training data remains a central challenge in modern machine learning (Shalev-Shwartz & Ben-David, 2014). While classical complexity measures often fail to provide tight bounds for deep networks, information-theoretic approaches offer a powerful alternative by linking generalization error to the information a learned hypothesis $W$ retains about the training data $S$. A seminal result by Xu & Raginsky (2017) formalized this, showing that for $\sigma^2$-subgaussian losses, the expected generalization gap is controlled by the mutual information (MI) $I(S; W)$.

This foundational work has inspired a rich literature aiming to refine and extend MI-based bounds. To tighten guarantees for deep networks, recent work has focused on "slicing" the mutual information to analyze per-layer or per-node dependencies (Nadjahi et al., 2024; Asadi et al., 2018). Others have developed techniques to achieve faster, $\mathcal{O}(1/n)$ rates under specific assumptions (Wang et al., 2023) or have used supersampling techniques to improve tightness (Bu et al., 2020). A key extension is the use of Conditional Mutual Information (CMI) (Steinke & Zakynthinou, 2020), which often yields stronger, high-probability guarantees. Despite these strengths, MI-based bounds exhibit notable limitations, such as looseness in high dimensions and a lack of adaptability, which motivate the exploration of alternative measures (Haghifam et al., 2023).

Recognizing these limitations, many have explored other divergences and metrics. Rényi divergences, which are closely related to our work, have been used to derive generalization bounds (Esposito et al., 2019) and have found applications in PAC-Bayesian frameworks (Guan et al., 2025). Aminian et al. (2022) specifically leveraged the convexity of information measures, including Rényi-type quantities, to tighten expected error bounds. More broadly, families like $f$-divergences (Liese & Vajda, 2006) and Integral Probability Metrics (IPMs) (M"uller, 1997; Sriperumbudur et al., 2012) provide general tools for bounding generalization. In parallel, Wasserstein distances from optimal transport theory have been successfully used to analyze the stability and generalization of gradient-based optimizers (Lopez & Jog, 2018; Rodríguez-Gálvez et al., 2021; Zhu et al., 2024).

Complementary to these information-theoretic views are approaches rooted in algorithmic stability. The foundational concept of uniform stability (Bousquet & Elisseeff, 2002) has led to highly refined,

sharp bounds for stable algorithms like SGD (Bousquet et al., 2020; Feldman & Vondr'ak, 2019). For stochastic algorithms, a veritable "zoo" of Bayesian stability notions has been cataloged, highlighting the diversity of approaches in this space (Moran et al., 2023). This existing work provides a backdrop for the variance-based perspective that emerges from our framework, a perspective that connects to other recent studies exploring the role of variance in variational inference (Wei et al., 2025), deep architectures (Li et al., 2025), and margin distributions (Chuang et al., 2021).

Several recent works have aimed to consolidate these disparate approaches into unifying frameworks (Raginsky et al., 2023; Haghifam et al., 2021). Most notably, Chu & Raginsky (2023) presented a highly general unification using the abstract language of Orlicz spaces. While these frameworks achieve a high degree of mathematical generality, our work offers a complementary unification. By leveraging the $V_\alpha$ measure, we provide a concrete framework that uses a single, intuitive parameter, $\alpha$, to smoothly interpolate between specific, well-understood measures—total variation, mutual information, and variance.

**Contributions.** This paper introduces a unified framework for deriving generalization error bounds using the tunable correlation measure $V_\alpha(S; W)$. Our primary contributions are:

1. **A Unified Information-Theoretic Framework:** We derive a one-parameter family of generalization bounds that, by tuning $\alpha$, recovers and interpolates between guarantees based on mutual information ($\alpha = 1$) and total variation ($\alpha \to \infty$).

2. **A New Variance-Based Perspective on Stability:** The framework's key conceptual insight emerges at $\alpha = 2$, yielding a novel bound that establishes the algorithm's output variance, $\text{Var}_S[p(w|S)]$, as a direct, data-dependent measure of its stability, providing a new information-theoretic explanation for how unstable (high-variance) algorithms fail to generalize.

3. **A Sufficient Condition for Non-Vacuous Generalization:** To demonstrate the utility of our variance-based bound, we introduce *Adaptive Density Stability*, a novel pointwise stability condition, and prove that algorithms satisfying it achieve non-vacuous generalization rates within our framework.

4. **Stronger Empirical Guarantees:** We validate our framework on a Bayesian linear regression task, showing that our $V_2$-based bound is empirically tighter than both classical MI and contemporary Conditional Mutual Information (CMI) bounds (Steinke & Zakynthinou, 2020).

The rest of the paper is organized as follows. Section 2 establishes the foundational preliminaries, including the standard statistical learning setup, a survey of key algorithmic stability notions, the definition of correlation measure $V_\alpha$, and a proof of its convexity. Section 3 details our core theoretical contributions: Subsection 3.1 derives the primary generalization bound in terms of $V_\alpha$ (Theorem 3.1), while Subsection 3.2 provides bounds on the associated $L_p$-norm term. Section 4 delves into specific instantiations of $\alpha$, recovering the mutual-information bound at $\alpha = 1$, a novel variance-based bound at $\alpha = 2$, and the worst-case total-variation bound as $\alpha \to \infty$; these analyses illuminate the inherent trade-offs and flexibility of our framework. Section 5 introduces adaptive density stability, a new distribution-centric measure that directly ties output variability to generalization performance. Section 6 empirically validates our approach through comprehensive simulations, comparing $V_\alpha$-based bounds against classical mutual-information baselines across diverse canonical channels and noisy classification tasks. Finally, Section 7 summarizes key implications and outlines avenues for future work.

## 2 PRELIMINARIES

In this section, we establish the foundational concepts from statistical learning theory and introduce the $V_\alpha$ correlation measure that is central to our framework.

### 2.1 STATISTICAL LEARNING SETUP

We follow the standard statistical learning framework. Let $\mathcal{Z} = \mathcal{X} \times \mathcal{Y}$ be the instance space and $\mathcal{W}$ be the hypothesis space. A learning algorithm is a procedure that maps a training sample

$S = (Z_1, \ldots, Z_n)$, drawn i.i.d. from an unknown data-generating distribution $\mu$, to a hypothesis $W \in \mathcal{W}$. This mapping is often randomized and is formally described by a conditional probability distribution $P_{W|S}$. The joint distribution over hypotheses and samples is thus $P_{W,S} = \mu^{\otimes n} \otimes P_{W|S}$.

The performance of a hypothesis $w \in \mathcal{W}$ is measured by its population risk, defined with respect to a loss function $\ell(w, z)$:

$$L_\mu(w) := \mathbb{E}_{Z \sim \mu}[\ell(w, Z)] = \int_{\mathcal{Z}} \ell(w, z) \mu(dz). \tag{1}$$

Since $\mu$ is unknown, an algorithm typically minimizes the empirical risk on the training set $S$:

$$L_S(w) := \frac{1}{n} \sum_{i=1}^{n} \ell(w, Z_i). \tag{2}$$

The key challenge is to ensure that a low empirical risk translates to a low population risk. This is captured by the generalization gap, defined for a learned hypothesis $W$ as $\text{gen}(W, S) := L_\mu(W) - L_S(W)$. We are interested in its expected value, the generalization error:

$$\overline{\text{gen}}\left(\mu, P_{W|S}\right) := \mathbb{E}_{P_{W,S}}\left[L_\mu(W) - L_S(W)\right]. \tag{3}$$

The expected population risk can then be decomposed as:

$$\mathbb{E}\left[L_\mu(W)\right] = \mathbb{E}\left[L_S(W)\right] + \overline{\text{gen}}\left(\mu, P_{W|S}\right). \tag{4}$$

This decomposition reveals the fundamental trade-off in machine learning: an algorithm must achieve a low empirical risk on the training data while also maintaining a small generalization error.

**Algorithmic Stability.** The concept of stability formalizes the intuition that algorithms that are insensitive to small changes in the training data tend to generalize well. Prominent notions include *uniform stability*, which bounds the worst-case change in loss from replacing a single data point (Bousquet & Elisseeff, 2002), and various *information-theoretic stability* measures that use divergences to quantify the sensitivity of the output distribution $P_{W|S}$ (Dwork et al., 2015; Raginsky et al., 2016).

**Subgaussian Random Variables.** Our analysis relies on the notion of subgaussianity, which characterizes random variables with tails at least as light as a Gaussian.

**Definition 2.1** (Vershynin (2018)). *A random variable $X$ is $\sigma^2$-subgaussian if for all $\lambda \in \mathbb{R}$, its moment generating function satisfies $\mathbb{E}\left[e^{\lambda(X - \mathbb{E}[X])}\right] \leq e^{\frac{\lambda^2 \sigma^2}{2}}$. This implies a tail bound $\mathbb{P}[|X - \mathbb{E}[X]| > t] \leq 2e^{-t^2/(2\sigma^2)}$ and a variance bound $\text{Var}[X] \leq \sigma^2$. A key result, Hoeffding's Lemma, states that if a random variable is bounded in $[a, b]$, it is $\frac{(b-a)^2}{4}$-subgaussian.*

**Definition 2.2.** *The $L_\alpha$-norm of a random variable $X$ is defined as*

$$\|X\|_\alpha = \left(\mathbb{E}\left[|X|^\alpha\right]\right)^{\frac{1}{\alpha}}. \tag{5}$$

*The $L_\alpha$-norm is an increasing function of $\alpha$, and for $\alpha > 1$, the $L_\alpha$-norm is convex in $X$.*

In Xu & Raginsky (2017), mutual information of $W$ and $S$, is related to the generalization error.

**Theorem 2.1** (Generalization error and stability (Xu & Raginsky, 2017)). *Suppose $\ell(w, Z)$ is $\sigma^2$-subgaussian under $Z \sim \mu$ for all $w \in \mathcal{W}$, then the generalization error is bounded as*

$$\left|\mathbb{E}_{P_{W,S}}\left[\text{gen}(\mu, P_{W|S})\right]\right| \leq \sqrt{\frac{2\sigma^2}{n} I(W; S)}. \tag{6}$$

## 2.2 THE $V_\alpha$ CORRELATION MEASURE

Our framework is built upon the vector-valued $L_p$-norm correlation measure $V_\alpha(A; B)$, introduced by Mojahedian et al. (2019). For a joint distribution $p_{AB}$, the measure is defined for $\alpha \geq 1$ as:

$$V_\alpha(A; B) := \mathbb{E}_{b \sim p_B}\left[\left(\mathbb{E}_{a \sim p_A}\left|\frac{p_{B|A}(b|a)}{p_B(b)} - 1\right|^\alpha\right)^{1/\alpha}\right]. \tag{7}$$

The measure is non-decreasing in $\alpha$. It provides a tunable bridge between several well-known dependence measures. For $\alpha = 1$, it recovers the total variation distance between the joint and product distributions: $V_1(A; B) = \|p_{AB} - p_A p_B\|_1$. It is also closely related to the Rényi mutual information of order $\alpha$, $I_\alpha(A; B)$, via the inequality (Mojahedian et al., 2019, Prop. 10):

$$2^{\frac{1}{\alpha'} I_\alpha(A;B)} - 1 \leq V_\alpha(A; B) \leq 2^{\frac{1}{\alpha'} I_\alpha(A;B)} + 1, \tag{8}$$

where $\alpha'$ is the Hölder conjugate of $\alpha$. This property allows $V_\alpha$ to smoothly interpolate from a linear dependence measure (total variation) to an exponential one (related to Rényi information).

### 2.3 CONVEXITY OF $V_\alpha$

A key property of the $V_\alpha$ measure, which is central to its analytical tractability, is its convexity. For discrete spaces, we can express $V_\alpha$ as a sum over the outcomes of $B$:

$$V_\alpha(A; B) = \sum_{b \in \mathcal{B}} \left( \mathbb{E}_{a \sim p_A} \left[ |p_{B|A}(b|a) - p_B(b)|^\alpha \right] \right)^{1/\alpha} = \sum_{b \in \mathcal{B}} \|p_{B|A}(b|\cdot) - p_B(b)\|_{\alpha, p_A}, \tag{9}$$

where $\|\cdot\|_{\alpha, p_A}$ denotes the $L_\alpha$-norm with respect to the measure $p_A$. Since the $L_\alpha$-norm is convex for $\alpha \geq 1$, and sums of convex functions are convex, $V_\alpha(A; B)$ is a convex function of the conditional distributions $p_{B|A}(\cdot|a)$ and the marginal distribution $p_B$. This property holds analogously for continuous spaces, where the sum is replaced by an integral. The convexity of $V_\alpha$ enables its use in optimization-based interpretations and simplifies its analysis.

## 3 THE $V_\alpha$-INFORMATION BOUND

Having established the necessary preliminaries, we now present our main theoretical results. We first introduce a general bound on the expected generalization error in terms of the $V_\alpha$ correlation measure and then provide a more concrete corollary under the subgaussian loss assumption.

### 3.1 A GENERAL BOUND ON THE GENERALIZATION ERROR

Our main theorem provides a flexible, one-parameter family of bounds on the generalization error. The proof, which relies on a standard application of Hölder's inequality, is provided in Appendix B.

**Theorem 3.1.** *For any learning algorithm and any loss function, the expected generalization error is bounded for every $\alpha \geq 1$ as:*

$$\overline{\text{gen}}\left(\mu, P_{W|S}\right) \leq \sup_{w \in \mathcal{W}} \|L_S(w) - L_\mu(w)\|_{\alpha'} \cdot V_\alpha(S; W), \tag{10}$$

*where $\alpha'$ denotes the Hölder conjugate of $\alpha$.*

**Remark** (Symmetry of the Bound). *The bound can also be expressed in terms of $V_\alpha(W; S)$ by swapping the roles of $S$ and $W$ in the derivation (see Appendix C). This allows the bound to be tightened by taking the minimum of the two forms:*

$$\overline{\text{gen}}\left(\mu, P_{W|S}\right) \leq \min \left\{ \sup_{w \in \mathcal{W}} \|L_S(w) - L_\mu(w)\|_{\alpha'} \cdot V_\alpha(S; W), \right.$$

$$\left. \sup_{s \in \mathcal{Z}^n} \|L_s(W) - L_\mu(W)\|_{\alpha'} \cdot V_\alpha(W; S) \right\}. \tag{11}$$

**Remark** (The Trade-off in $\alpha$). *The bound in Theorem 3.1 reveals a fundamental trade-off controlled by the parameter $\alpha$. The information term, $V_\alpha(S; W)$, is a non-decreasing function of $\alpha$ (Mojahedian et al., 2019). In contrast, the loss sensitivity term, $\|L_S(w) - L_\mu(w)\|_{\alpha'}$, is an $L_p$-norm of a random variable, which is non-increasing in $\alpha$. These opposing trends suggest that the bound can be tightened by optimizing over $\alpha$ to find the optimal balance for a given problem.*

*To illustrate this trade-off, consider a simple Z-channel model where the channel input $S$ is a single sample from a Bernoulli(q) distribution, and the output is the hypothesis $W$. Let the loss be the squared error $\ell(w, z) = (w - z)^2$. In Figure 1, we plot the two components of our bound and their*

*product as a function of $\alpha$. The information term ($V_\alpha$) grows with $\alpha$, while the loss sensitivity term ($\|\cdot\|_{\alpha'}$) shrinks. Their product, our bound, achieves a minimum, in this case at $\alpha = 2$, where it becomes tightest. For comparison, we also plot the classical mutual information bound, which is looser across the entire range.*

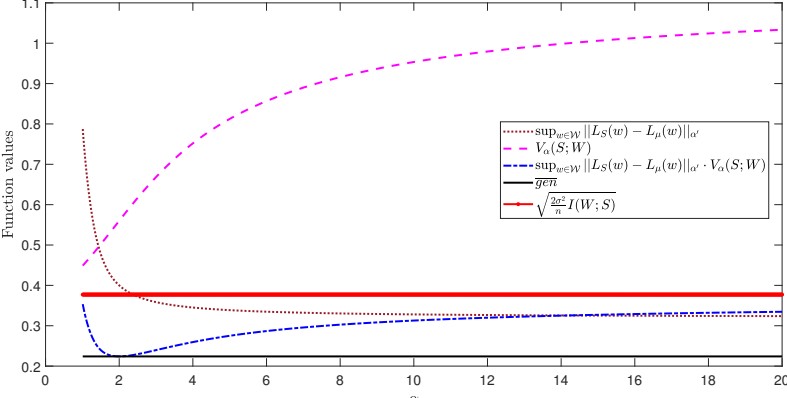

Figure 1: Illustration of the trade-off in Theorem 3.1. As $\alpha$ increases, the information term $V_\alpha(S; W)$ grows while the loss sensitivity term $\|\cdot\|_{\alpha'}$ shrinks. Their product (our bound) is minimized at an intermediate value, providing a tighter guarantee than the classical mutual information bound.

### 3.2 A Bound for Subgaussian Losses

While Theorem 3.1 is general, we can derive a more concrete and directly comparable bound by assuming the loss is subgaussian. This allows us to bound the loss sensitivity term, yielding the following corollary. The proof is provided in Appendix D.

**Corollary 3.1.** *Suppose that the loss $\ell(w, Z)$ is $\sigma^2$-subgaussian under $Z \sim \mu$ for all $w \in \mathcal{W}$. Then, for any $\alpha \geq 1$, the generalization error satisfies:*

$$\overline{\text{gen}}\left(\mu, P_{W|S}\right) \leq \left(\sqrt{\frac{2\sigma^2}{n}}(\alpha')^{1/\alpha'}\Gamma\left(\frac{\alpha'}{2}\right)^{1/\alpha'}\right) V_\alpha(S; W), \tag{12}$$

*where $\Gamma(\cdot)$ is the Gamma function. For large $\alpha'$, this can be further simplified to $\mathcal{O}\left(\sqrt{\frac{\sigma^2\alpha'}{n}}\right) V_\alpha(S; W)$.*

## 4 Special Cases of $\alpha$

The true power of the $V_\alpha$ framework lies in its ability to interpolate between different types of generalization guarantees by tuning the parameter $\alpha$. In this section, we analyze the three most illustrative special cases: $\alpha = 1$ (mutual information), $\alpha = 2$ (variance and stability), and $\alpha \to \infty$ (worst-case deviation).

### 4.1 The $\alpha = 1$ Case: Recovering the Total Variation and Mutual Information Bounds

For $\alpha = 1$, the Hölder conjugate is $\alpha' = \infty$. In this regime, the $L_{\alpha'}$-norm becomes the essential supremum, and our bound from Theorem 3.1 takes the form:

$$\overline{\text{gen}}\left(\mu, P_{W|S}\right) \leq \sup_{w \in \mathcal{W}} \left\{\text{ess sup}_{S \sim \mu^{\otimes n}} |L_S(w) - L_\mu(w)|\right\} \cdot V_1(S; W). \tag{13}$$

As noted in Section 2, $V_1(S; W)$ is precisely the total variation distance $\|P_{W,S} - P_W P_S\|_1$. By applying Pinsker's inequality, which states that $\|P - Q\|_1 \leq \sqrt{2D_{\mathrm{KL}}(P\|Q)}$, we can further bound the total variation term by the mutual information:

$$\overline{\mathrm{gen}}\left(\mu, P_{W|S}\right) \leq \sup_{w \in \mathcal{W}} \left\{\mathrm{ess\,sup}_S |L_S(w) - L_\mu(w)|\right\} \cdot \sqrt{2I(S; W)}. \tag{14}$$

This recovers a bound that, like the classical result of Xu & Raginsky (2017), depends on the square root of the mutual information, but is weighted by a worst-case deviation of the empirical risk rather than a subgaussian constant.

### 4.2 The $\alpha = 2$ Case: A Variance-Based Perspective on Algorithmic Stability

The case of $\alpha = 2$ is the most conceptually novel outcome of our framework. Here, the Hölder conjugate is $\alpha' = 2$, and the $L_2$-norm of a zero-mean random variable is its standard deviation. This specialization yields a direct and intuitive connection between generalization and the variance of the learning algorithm's output distribution.

**Theorem 4.1.** *Suppose the loss $\ell(w, Z)$ is $\sigma^2$-subgaussian under $Z \sim \mu$ for all $w \in \mathcal{W}$. Then the generalization error is bounded as:*

$$\overline{\mathrm{gen}}\left(\mu, P_{W|S}\right) \leq \sum_{w \in \mathcal{W}} \sqrt{\frac{\sigma^2}{n} \mathrm{Var}_S\left[p(w|S)\right]}. \tag{15}$$

*Proof Sketch.* For $\alpha = 2$, Corollary 3.1 simplifies, giving $\overline{\mathrm{gen}} \leq \sqrt{\frac{\sigma^2}{n}} V_2(S; W)$. The theorem follows by applying an identity from Mojahedian et al. (2019), which states that $V_2(S; W) = \sum_w \sqrt{\mathrm{Var}_S[p(w|S)]}$. The full proof is in Appendix E. For continuous hypothesis spaces, the sum is replaced by an integral, provided it is well-defined. $\qquad\square$

The bound in Theorem 4.1 reveals a direct link between generalization and algorithmic stability. Each term $\mathrm{Var}_S[p(w|S)]$ quantifies the variability of the algorithm's output distribution across different training samples, serving as a data-dependent measure of its stability. High variance implies sensitivity to data resampling, a hallmark of overfitting, which loosens the bound. Conversely, low variance implies robustness, leading to a tighter guarantee.

**Relationship to Classical Stability.** We emphasize that this variance-based notion of stability is distinct from classical deterministic measures like uniform stability (Bousquet & Elisseeff, 2002). While uniform stability offers a worst-case guarantee on the change in loss from altering a single data point, our measure captures the on-average sensitivity of the entire output distribution over the data-generating process. Our framework does not recover or generalize these classical bounds. Instead, its contribution is to propose this variance as a new, direct measure of stability and to prove its fundamental connection to the generalization error.

**Comparison to Uniform Stability Bounds.** A natural question is how our variance-based bound compares to bounds based on uniform stability, such as the sharp results of Feldman & Vondr'ak (2019) or Bousquet et al. (2020). A direct analytical comparison is difficult, as the bounds depend on fundamentally different quantities. Uniform stability bounds depend on a worst-case, data-independent parameter $\gamma$, providing strong robustness guarantees. Our bound is data-dependent and average-case; it can be much tighter if an algorithm is stable on average, even if it is not strictly uniformly stable. The approaches are therefore complementary, with our framework offering an alternative perspective that can be more reflective of typical performance, as shown in our experiments in Section 6.

### 4.3 The $\alpha \to \infty$ Case: Worst-Case Deviation Bound

As $\alpha \to \infty$, its conjugate $\alpha' \to 1$. The $L_1$-norm of the (zero-mean) risk deviation is $\mathbb{E}_S[|L_S(w) - L_\mu(w)|]$. The information term $V_\infty(S; W)$ becomes a measure of the maximum pointwise deviation

of the conditional from the marginal:

$$V_\infty(S; W) = \sum_{w \in \mathcal{W}} \sup_{s \in \mathcal{Z}^n} |p(w|s) - p(w)|. \tag{16}$$

The resulting bound is thus:

$$\overline{\text{gen}}\left(\mu, p_{W|S}\right) \leq \sup_{w \in \mathcal{W}} \{\mathbb{E}_S\left[|L_S(w) - L_\mu(w)|\right]\} \cdot V_\infty(S; W). \tag{17}$$

This bound captures a trade-off between the average absolute deviation of the loss and the worst-case deviation of the algorithm's output distribution.

## 5 ADAPTIVE DENSITY STABILITY: A SUFFICIENT CONDITION FOR NON-VACUOUS BOUNDS

The variance-based bound in Theorem 4.1 provides a powerful conceptual link between stability and generalization. However, a key question remains: under what formal conditions on a learning algorithm is the variance term $\text{Var}_S[p(w|S)]$ guaranteed to be small enough to yield a non-vacuous, decaying generalization bound? To address this, we introduce a novel, strong stability condition and show that it serves as a sufficient guarantee for our framework to produce meaningful generalization rates.

**Definition 5.1** (Adaptive Density Stability). *A learning algorithm is said to be $(\gamma, n)$-adaptively density stable if for all datasets $S, S'$ of size $n$ that differ in a single sample, and for all hypotheses $w \in \mathcal{W}$, the following holds:*

$$|p(w|S) - p(w|S')| \leq \gamma_n p(w). \tag{18}$$

*The stability parameter $\gamma_n$ may depend on the sample size $n$.*

This definition formalizes a notion of distributional stability where the allowed change in the output density at a point $w$ is relative to the marginal probability $p(w)$. This adaptively places a stronger stability constraint on the low-probability tails of the output distribution. As shown in Appendix F, this strong, pointwise condition implies both classical TV stability and Expected Uniform Stability.

Our main result in this section is to show that algorithms satisfying this condition have a controlled $V_2$ measure, which in turn bounds the generalization error.

**Theorem 5.1.** *If a learning algorithm is $(\gamma_n, n)$-adaptively density stable, then its $V_2$ correlation measure is bounded by:*

$$V_2(S; W) \leq \sqrt{2\gamma_n n}. \tag{19}$$

*Consequently, for a $\sigma^2$-subgaussian loss, the generalization error is bounded by:*

$$\overline{\text{gen}}\left(\mu, P_{W|S}\right) \leq \sqrt{2\sigma^2 \gamma_n}. \tag{20}$$

*Proof.* The proof, which involves an application of the Efron-Stein inequality, is provided in Appendix G. □

**On Non-Vacuousness.** At first glance, the bound $\sqrt{2\sigma^2\gamma_n}$ appears independent of $n$ and potentially vacuous. However, the utility of any stability-based framework hinges on the stability parameter itself improving as the sample size grows. For our bound to be meaningful, we require the stability parameter $\gamma_n$ to decrease with $n$.

This requirement is standard in the literature. For many algorithms, particularly those regularized by noise (e.g., Stochastic Gradient Langevin Dynamics), stability is achieved by setting noise levels or other parameters relative to the sample size. It is often possible to show that stability parameters analogous to ours can be made to scale as $\gamma_n = \mathcal{O}(1/n)$ (e.g., Raginsky et al., 2016). If an algorithm satisfies our condition with such a rate, our bound becomes non-vacuous and recovers a standard learning rate:

$$\overline{\text{gen}} \leq \sqrt{2\sigma^2 \cdot \mathcal{O}(1/n)} = \mathcal{O}\left(\frac{1}{\sqrt{n}}\right). \tag{21}$$

Thus, our framework is not inherently vacuous. Its practical utility is conditioned on verifying the decay rate of $\gamma_n$ for a given algorithm, which is the central task in applying any such generalization bound. While a full analysis for specific complex algorithms is outside the scope of this paper, Definition 5.1 provides a clear and sufficient condition for our variance-based theory to be predictive.

# 6 EXPERIMENTS

In this section, we empirically validate our theoretical framework. While our bounds hold for any $\alpha \geq 1$, we focus on the $\alpha = 2$ case, as the resulting variance-based bound (Theorem 4.1) represents our most significant conceptual contribution. We present a new experiment on a Bayesian linear regression task to demonstrate the bound's performance in a practical machine learning scenario, comparing it against strong contemporary baselines. Additional simulations on canonical channel models are provided in Appendix H.

## 6.1 SETUP: BAYESIAN LINEAR REGRESSION

**Learning Task.** We consider a one-dimensional Bayesian linear regression task where data is generated as $y = w_{\text{true}}x + \epsilon$, with $x \sim \mathcal{N}(0,1)$ and i.i.d. Gaussian noise $\epsilon \sim \mathcal{N}(0,\sigma^2)$. To ensure the loss is $\sigma_{\text{loss}}^2$-subgaussian, as required by our theory, we use a clipped squared error loss: $\ell(w,z) = \min\{(y - wx)^2, c\}$, where $c > 0$ is a fixed clipping constant.

**Learning Algorithm.** The algorithm is a standard Bayesian linear regression with a Gaussian prior on the weight, $p(w) = \mathcal{N}(0, \lambda^{-1})$, where $\lambda$ is the prior precision. Due to conjugacy, the posterior distribution $p(w|S)$ is also a Gaussian, which allows for the analytical computation of all quantities required for the bounds.

## 6.2 RESULTS: COMPARISON WITH INFORMATION-THEORETIC BASELINES

We evaluate our proposed $V_2$ bound from Theorem 4.1 against three key quantities: the empirically estimated true generalization error, the classical mutual information (MI) bound of Xu & Raginsky (2017), and the stronger, contemporary Conditional Mutual Information (CMI) bound of Steinke & Zakynthinou (2020). All quantities are estimated via Monte Carlo simulation across multiple training sets of varying sizes $n$. Full implementation details are provided in Appendix H.

The results are presented in Figure 2. As expected, the CMI bound is consistently tighter than the classical MI bound. Our key finding is that the $V_2$ bound is even tighter than the CMI bound across all sample sizes, providing a sharp and valid upper bound that closely tracks the true generalization error. This result demonstrates that the variance-based perspective offered by our framework can yield not only conceptual insights but also state-of-the-art empirical guarantees.

# 7 CONCLUSION AND FUTURE WORK

In this work, we introduced a flexible, one-parameter family of information-theoretic generalization bounds based on the $V_\alpha$ correlation measure. Our framework provides a unified lens through which to view and interpolate between guarantees based on mutual information, total variation, and variance. The primary conceptual contribution of our work emerged at the $\alpha = 2$ specialization, which yielded a novel and intuitive variance-based bound. This result establishes the variance of an algorithm's output distribution, $\text{Var}_S[p(w|S)]$, as a direct, data-dependent measure of its stability, proving that this quantity directly controls the generalization error. Our empirical validation on a Bayesian linear regression task demonstrated that this new perspective is not only theoretically insightful but can also yield generalization bounds that are significantly tighter than strong contemporary baselines like Conditional Mutual Information.

While our theoretical results are general, a key limitation of our current empirical validation is its focus on models where the relevant information-theoretic quantities are analytically tractable. This is a common challenge in the field, as estimating measures like $I(S;W)$ or $V_\alpha(S;W)$ for high-dimensional models such as deep neural networks remains a significant computational hurdle. This practical consideration motivates several important directions for future research.

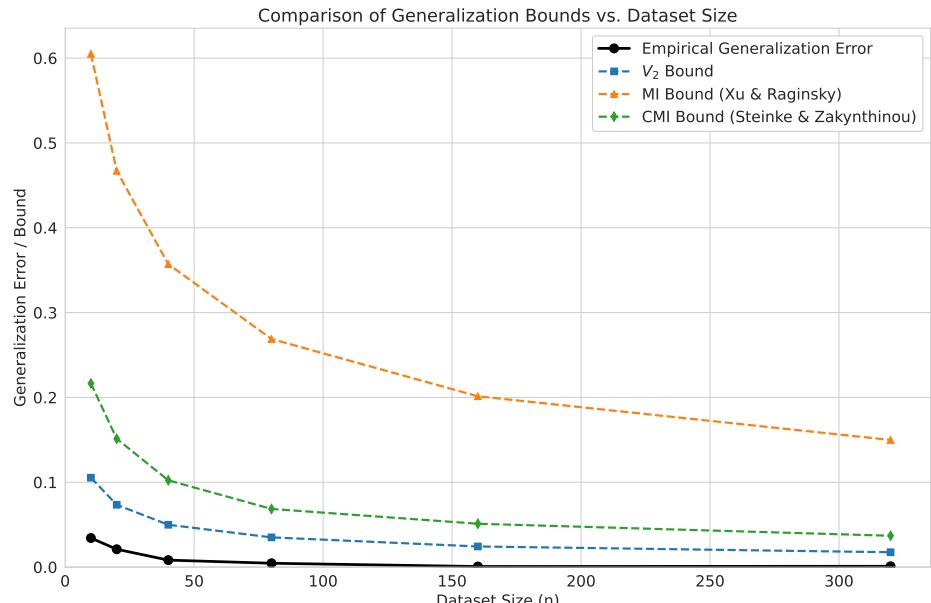

Figure 2: Comparison of generalization bounds in Bayesian linear regression. The plot shows the true generalization error (estimated empirically) against the MI bound (Xu & Raginsky, 2017), the CMI bound (Steinke & Zakynthinou, 2020), and our proposed $V_2$ variance-based bound (Theorem 4.1). The $V_2$ bound provides the tightest upper bound on the generalization error across all sample sizes.

Future work could proceed along the following lines:

- **Practical Estimation for Deep Networks:** Developing scalable and reliable estimators for the $V_\alpha(S; W)$ measure for deep neural networks. This would enable the application of our bounds to modern, overparameterized architectures and allow for a direct test of their utility on large-scale datasets.

- **Extending the Theoretical Framework:** Broadening our analysis to derive high-probability bounds under weaker assumptions and extending the framework to handle non-subgaussian or heavy-tailed loss distributions.

- **Analysis via Adaptive Density Stability:** A key theoretical direction is to formally analyze which classes of algorithms satisfy strong stability conditions, such as the Adaptive Density Stability we introduced, in order to provide a priori guarantees that the variance term in our bound will be small.

- **Adaptive Selection of $\alpha$:** Developing principled methods for adaptively selecting the optimal value of $\alpha$ for a given learning problem, which would allow practitioners to obtain the tightest possible bound from our framework.

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

# A  LLM USAGE

A Large Language Model (LLM) was used as a writing and editing assistant during the preparation of this manuscript. Its role was limited to improving clarity, flow, and structure, including polishing sentences, suggesting concise alternatives, providing feedback on section organization, refining the articulation of key concepts, and assisting with LaTeX formatting. All research ideas, mathematical derivations, experimental design, and scientific claims are solely the work of the human authors, who take full responsibility for the final content of the paper.

# B  PROOF OF THEOREM 3.1

The expected generalization error is defined as:

$$\overline{\text{gen}}\left(\mu, P_{W|S}\right) = \mathbb{E}_{P_{W,S}}\left[L_\mu(W) - L_S(W)\right]. \tag{22}$$

We can rewrite this expectation by first conditioning on $W$:

$$\overline{\text{gen}}\left(\mu, P_{W|S}\right) = \mathbb{E}_{W \sim P_W}\left[\mathbb{E}_{S \sim P_{S|W}}\left[L_\mu(W) - L_S(W)\right]\right]. \tag{23}$$

The core of the proof relies on the following identity, which connects the conditional expectation over $P_{S|W}$ to an expectation over the marginal $P_S$:

$$\mathbb{E}_{S \sim P_{S|W}}\left[L_\mu(W) - L_S(W)\right] = \mathbb{E}_{S \sim P_S}\left[(L_\mu(W) - L_S(W))\left(\frac{p_{S|W}(S|W)}{p_S(S)}\right)\right]. \tag{24}$$

This holds because for any function $f(S)$, $\mathbb{E}_{S \sim P_{S|W}}[f(S)] = \int f(s) p_{S|W}(s|W) ds = \int f(s)\frac{p_{S|W}(s|W)}{p_S(s)} p_S(s) ds$.

Furthermore, we note that $\mathbb{E}_{S \sim P_S}[L_S(W)] = L_\mu(W)$, since:

$$\mathbb{E}_{S \sim \mu^{\otimes n}}\left[L_S(W)\right] = \mathbb{E}_{S \sim \mu^{\otimes n}}\left[\frac{1}{n}\sum_{i=1}^{n}\ell(W, Z_i)\right] = \frac{1}{n}\sum_{i=1}^{n}\mathbb{E}_{Z_i \sim \mu}\left[\ell(W, Z_i)\right] = L_\mu(W). \tag{25}$$

This implies that $\mathbb{E}_{S \sim P_S}[L_\mu(W) - L_S(W)] = 0$. Using this, we can rewrite the identity in equation 24 as:

$$\mathbb{E}_{S \sim P_{S|W}}\left[L_\mu(W) - L_S(W)\right] = \mathbb{E}_{S \sim P_S}\left[(L_\mu(W) - L_S(W))\left(\frac{p_{S|W}(S|W)}{p_S(S)} - 1\right)\right]. \tag{26}$$

Substituting this back into the expression for the generalization error gives:

$$\overline{\text{gen}} = \mathbb{E}_{W \sim P_W}\left[\mathbb{E}_{S \sim P_S}\left[(L_\mu(W) - L_S(W))\left(\frac{p_{S|W}(S|W)}{p_S(S)} - 1\right)\right]\right] \tag{27}$$

$$\leq \mathbb{E}_{W \sim P_W}\left[\mathbb{E}_{S \sim P_S}\left[|L_\mu(W) - L_S(W)| \cdot \left|\frac{p_{S|W}(S|W)}{p_S(S)} - 1\right|\right]\right] \tag{28}$$

We now apply Hölder's inequality to the inner expectation over $S$ with conjugate exponents $\alpha'$ and $\alpha$:

$$\overline{\text{gen}} \leq \mathbb{E}_{W \sim P_W}\left[\|L_\mu(W) - L_S(W)\|_{\alpha'} \cdot \left\|\frac{p_{S|W}(S|W)}{p_S(S)} - 1\right\|_\alpha\right] \tag{29}$$

$$= \mathbb{E}_{W \sim P_W}\left[\|L_S(W) - L_\mu(W)\|_{\alpha'} \cdot \left(\mathbb{E}_{S \sim P_S}\left|\frac{p_{W|S}(W|S)p_S(S)}{p_W(W)p_S(S)} - 1\right|^\alpha\right)^{1/\alpha}\right] \tag{30}$$

where in equation 30 we used Bayes' rule, $p_{S|W} = p_{W|S}p_S/p_W$, in the second term.

We can now pull the supremum of the loss sensitivity term out of the expectation over $W$:

$$\overline{\text{gen}} \leq \sup_{w \in \mathcal{W}}\left\{\|L_S(w) - L_\mu(w)\|_{\alpha'}\right\} \cdot \mathbb{E}_{W \sim P_W}\left[\left(\mathbb{E}_{S \sim P_S}\left|\frac{p_{W|S}(W|S)}{p_W(W)} - 1\right|^\alpha\right)^{1/\alpha}\right] \tag{31}$$

$$= \sup_{w \in \mathcal{W}}\left\{\|L_S(w) - L_\mu(w)\|_{\alpha'}\right\} \cdot V_\alpha(S; W), \tag{32}$$

where the final step uses the definition of $V_\alpha(S; W)$ from equation 7. This completes the proof.

## C  PROOF OF REMARK 3.1

The proof of the alternative form of the bound begins from equation 28 in the proof of Theorem 3.1. Instead of conditioning on $W$ first, we can write the total expectation by conditioning on $S$ first:

$$\overline{\text{gen}}\left(\mu, P_{W|S}\right) \leq \mathbb{E}_{W \sim P_W}\left[\mathbb{E}_{S \sim P_S}\left[|L_\mu(W) - L_S(W)| \cdot \left|\frac{p_{S|W}(S|W)}{p_S(S)} - 1\right|\right]\right] \tag{33}$$

$$= \mathbb{E}_{S \sim P_S}\left[\mathbb{E}_{W \sim P_W}\left[|L_S(W) - L_\mu(W)| \cdot \left|\frac{p_{S|W}(S|W)}{p_S(S)} - 1\right|\right]\right]. \tag{34}$$

We now apply Hölder's inequality to the inner expectation over $W$ with conjugate exponents $\alpha'$ and $\alpha$:

$$\overline{\text{gen}} \leq \mathbb{E}_{S \sim P_S}\left[\|L_S(W) - L_\mu(W)\|_{\alpha'} \cdot \left\|\frac{p_{S|W}(S|W)}{p_S(S)} - 1\right\|_\alpha\right] \tag{35}$$

$$= \mathbb{E}_{S \sim P_S}\left[\|L_S(W) - L_\mu(W)\|_{\alpha'} \cdot \left(\mathbb{E}_{W \sim P_W}\left|\frac{p_{S|W}(S|W)}{p_S(S)} - 1\right|^\alpha\right)^{1/\alpha}\right] \tag{36}$$

We can now pull the supremum of the loss sensitivity term out of the expectation over $S$:

$$\overline{\text{gen}} \leq \sup_{s \in \mathcal{Z}^n}\left\{\|L_s(W) - L_\mu(W)\|_{\alpha'}\right\} \cdot \mathbb{E}_{S \sim P_S}\left[\left(\mathbb{E}_{W \sim P_W}\left|\frac{p_{S|W}(s|W)}{p_S(s)} - 1\right|^\alpha\right)^{1/\alpha}\right] \tag{37}$$

$$= \sup_{s \in \mathcal{Z}^n}\left\{\|L_s(W) - L_\mu(W)\|_{\alpha'}\right\} \cdot V_\alpha(W; S). \tag{38}$$

The final, tightened bound is obtained by taking the minimum of this result and the one derived in Theorem 3.1.

## D  PROOF OF COROLLARY 3.1

The proof of Corollary 3.1 relies on combining the general bound from Theorem 3.1 with a standard result that bounds the $L_p$-norm of a subgaussian random variable.

We begin by stating the necessary lemma, which bounds the moments of a subgaussian random variable.

**Lemma D.1** (see, e.g., Rivasplata (2012), Proposition 3.2). *If a random variable $X$ is $\sigma^2$-subgaussian, then for any $p \geq 1$, its $L_p$-norm is bounded as:*

$$\|X\|_p = \left(\mathbb{E}|X|^p\right)^{1/p} \leq \sqrt{2\sigma^2}p^{1/p}\Gamma\left(\frac{p}{2}\right)^{1/p}, \tag{39}$$

*where $\Gamma(\cdot)$ is the Gamma function.*

Now, consider the loss sensitivity term in Theorem 3.1: $\sup_{w \in \mathcal{W}}\|L_S(w) - L_\mu(w)\|_{\alpha'}$. The term inside the norm, $L_S(w) - L_\mu(w)$, is the difference between an empirical mean of $n$ i.i.d. random variables and its expectation. By the assumption of Corollary 3.1, the loss $\ell(w, Z)$ is $\sigma^2$-subgaussian for any fixed $w$. A standard property of subgaussian variables is that their sum is also subgaussian, with the variance parameter scaling accordingly. Therefore, the empirical risk $L_S(w) = \frac{1}{n}\sum_{i=1}^n \ell(w, Z_i)$ is a $\frac{\sigma^2}{n}$-subgaussian random variable.

This implies that the zero-mean random variable $X_w = L_S(w) - L_\mu(w)$ is $\frac{\sigma^2}{n}$-subgaussian. We can now apply Lemma D.1 with $p = \alpha'$ and variance parameter $\frac{\sigma^2}{n}$ to bound its $L_{\alpha'}$-norm:

$$\|L_S(w) - L_\mu(w)\|_{\alpha'} \leq \sqrt{2 \cdot \frac{\sigma^2}{n}}(\alpha')^{1/\alpha'}\Gamma\left(\frac{\alpha'}{2}\right)^{1/\alpha'}. \tag{40}$$

Since this bound holds uniformly for all $w \in \mathcal{W}$, it also holds for the supremum over $\mathcal{W}$.

Finally, substituting this inequality back into the main bound from Theorem 3.1 yields the desired result:

$$\overline{\text{gen}}\left(\mu, P_{W|S}\right) \le \sup_{w \in \mathcal{W}} \left\{\|L_S(w) - L_\mu(w)\|_{\alpha'}\right\} \cdot V_\alpha(S; W) \tag{41}$$

$$\le \left(\sqrt{\frac{2\sigma^2}{n}}(\alpha')^{1/\alpha'}\Gamma\left(\frac{\alpha'}{2}\right)^{1/\alpha'}\right) V_\alpha(S; W). \tag{42}$$

This completes the proof.

## E    PROOF OF THEOREM 4.1

The variance-based bound in Theorem 4.1 is a specialization of our main result, Theorem 3.1, for the case of $\alpha = 2$. The proof proceeds in three steps.

**Step 1: Specialize the general bound to $\alpha = 2$.**    We begin with the general bound from Theorem 3.1:

$$\overline{\text{gen}}\left(\mu, P_{W|S}\right) \le \sup_{w \in \mathcal{W}} \|L_S(w) - L_\mu(w)\|_{\alpha'} \cdot V_\alpha(S; W). \tag{43}$$

For $\alpha = 2$, the Hölder conjugate is $\alpha' = 2$. The $L_2$-norm of the zero-mean random variable $L_S(w) - L_\mu(w)$ is, by definition, its standard deviation:

$$\|L_S(w) - L_\mu(w)\|_2 = \left(\mathbb{E}_S\left[(L_S(w) - L_\mu(w))^2\right]\right)^{1/2} = \sqrt{\text{Var}_S[L_S(w)]}. \tag{44}$$

Substituting $\alpha = 2$ and this identity into the general bound gives:

$$\overline{\text{gen}}\left(\mu, P_{W|S}\right) \le \sup_{w \in \mathcal{W}} \left\{\sqrt{\text{Var}_S[L_S(w)]}\right\} \cdot V_2(S; W). \tag{45}$$

**Step 2: Apply the subgaussian assumption.**    The theorem assumes that the loss $\ell(w, Z)$ is $\sigma^2$-subgaussian for all $w$. As noted in the proof of Corollary 3.1, this implies that the empirical risk $L_S(w)$ is a $\frac{\sigma^2}{n}$-subgaussian random variable. A key property of subgaussian variables (see Definition 2.1) is that their variance is bounded by their subgaussian parameter. Therefore, we have:

$$\text{Var}_S[L_S(w)] \le \frac{\sigma^2}{n} \quad \text{for all } w \in \mathcal{W}. \tag{46}$$

Substituting this uniform bound into equation 45, we get:

$$\overline{\text{gen}}\left(\mu, P_{W|S}\right) \le \sqrt{\frac{\sigma^2}{n}} \cdot V_2(S; W). \tag{47}$$

**Step 3: Express $V_2(S; W)$ in terms of variance.**    The final step is to use an alternative identity for the $V_2$ measure provided by Mojahedian et al. (2019, Sec. III). For a discrete hypothesis space $\mathcal{W}$, this identity is:

$$V_2(S; W) = \sum_{w \in \mathcal{W}} \sqrt{\text{Var}_S[p(w|S)]}. \tag{48}$$

Substituting this identity into equation 47 yields the final result for the discrete case:

$$\overline{\text{gen}}\left(\mu, P_{W|S}\right) \le \sum_{w \in \mathcal{W}} \sqrt{\frac{\sigma^2}{n}\text{Var}_S[p(w|S)]}. \tag{49}$$

**Extension to Continuous Spaces.**    For a continuous hypothesis space, the sum in the identity for $V_2$ (equation 48) is replaced by an integral:

$$V_2(S; W) = \int_{\mathcal{W}} \sqrt{\text{Var}_S[p(w|S)]} \, dw, \tag{50}$$

where $p(w|S)$ is now the probability density function. The final bound in the theorem then takes the corresponding integral form, provided the integral is well-defined. This completes the proof.

## F    DEEPER ANALYSIS OF ADAPTIVE DENSITY STABILITY

In this section, we provide a more detailed analysis of the Adaptive Density Stability condition introduced in Definition 5.1. We demonstrate that despite its novelty, our definition is well-grounded within the broader literature on algorithmic stability by formally connecting it to several established concepts. These connections show that our condition is a strong and meaningful notion of distributional stability.

### F.1    CONNECTION TO TOTAL VARIATION (TV) STABILITY

Our stability notion is a strong, pointwise condition that directly implies the standard integrated notion of Total Variation (TV) stability. An algorithm is considered TV-stable if the TV distance between its output distributions on any two neighboring datasets $S$ and $S'$ is bounded. We show that our condition provides such a bound.

By integrating our stability condition from Definition 5.1 over the hypothesis space $\mathcal{W}$, we can bound the TV distance:

$$\|p(w|S) - p(w|S')\|_{\text{TV}} = \frac{1}{2} \int_{\mathcal{W}} |p(w|S) - p(w|S')| \; dw \tag{51}$$

$$\leq \frac{1}{2} \int_{\mathcal{W}} \gamma_n p(w) \, dw \quad \text{(by Definition 5.1)} \tag{52}$$

$$= \frac{\gamma_n}{2} \int_{\mathcal{W}} p(w) \, dw = \frac{\gamma_n}{2}, \tag{53}$$

since $p(w)$ is a probability density. This proves that an algorithm that is $(\gamma_n, n)$-adaptively density stable is also $\frac{\gamma_n}{2}$-TV stable. Our pointwise criterion is therefore strictly stronger than the integrated TV stability condition.

### F.2    CONNECTION TO EXPECTED UNIFORM STABILITY

Furthermore, our distributional stability implies the classical notion of stability for the expected loss. For a stochastic algorithm, Expected Uniform Stability is a natural extension of the definition from Bousquet & Elisseeff (2002). Assuming a loss function $\ell(w, z)$ is bounded by a constant $M$, an algorithm is $\beta$-stable if:

$$\sup_{z \in \mathcal{Z}} \left| \mathbb{E}_{W \sim p_{W|S}}[\ell(W, z)] - \mathbb{E}_{W' \sim p_{W|S'}}[\ell(W', z)] \right| \leq \beta. \tag{54}$$

We show that our stability notion implies this property:

$$\left| \mathbb{E}_{W \sim p_{W|S}}[\ell(W, z)] - \mathbb{E}_{W' \sim p_{W|S'}}[\ell(W', z)] \right| \tag{55}$$

$$= \left| \int_{\mathcal{W}} \ell(w, z)(p(w|S) - p(w|S')) \, dw \right| \tag{56}$$

$$\leq \int_{\mathcal{W}} |\ell(w, z)| \cdot |p(w|S) - p(w|S')| \, dw \tag{57}$$

$$\leq \int_{\mathcal{W}} M \cdot (\gamma_n p(w)) \, dw \quad \text{(using Definition 5.1 and } |\ell| \leq M) \tag{58}$$

$$= M \gamma_n \int_{\mathcal{W}} p(w) \, dw = M \gamma_n. \tag{59}$$

Thus, an algorithm satisfying our condition with parameter $\gamma_n$ also satisfies Expected Uniform Stability with parameter $\beta = M \gamma_n$. This highlights that our condition is more fundamental, as stability of the entire output distribution naturally implies the stability of an integrated property like its expected loss.

### F.3 COMPARISON WITH DIFFERENTIAL PRIVACY

It is also instructive to compare our stability notion to Differential Privacy (DP), another powerful framework for ensuring distributional stability. An algorithm satisfying pure $\epsilon$-DP adheres to:

$$p(w|S) \leq e^{\epsilon} p(w|S') \quad \forall w, S, S'.$$

This multiplicative, worst-case guarantee is different from our additive, marginal-relative guarantee. However, both enforce strong constraints on the output distribution. A known property of DP (often called group privacy) allows it to be extended to a bound relative to the marginal $p(w)$, yielding $\frac{p(w|S)}{p(w)} \leq e^{n\epsilon}$. This has a direct consequence on the term inside our $V_2$ measure, implying an upper bound on $V_2$ itself: $V_2(S; W) \leq e^{n\epsilon} - 1 \approx n\epsilon$ for small $\epsilon$. This suggests a deep connection between the stability required for privacy and the stability required for generalization in our framework.

### F.4 COMPARISON WITH BAYESIAN STABILITY

The framework of Bayesian stability, as surveyed in Moran et al. (2023), typically analyzes stability by measuring the dissimilarity $d(A(S), \mathcal{P})$ between the posterior $A(S)$ and a reference distribution $\mathcal{P}$ (often a prior), for example using the KL divergence. Our approach offers a complementary perspective. It is a **pairwise, leave-one-out style constraint**, akin to Differential Privacy, rather than a direct comparison to a fixed reference distribution.

The primary distinction is that our notion is a **pointwise** constraint on the absolute difference of the densities, whereas a measure like KL-stability is an **integrated** constraint. This makes our condition formally stronger in certain respects; it guarantees that for *every single hypothesis* $w$, the change in its probability density is controlled, forbidding large local deviations that an integrated measure like KL divergence might permit. This pointwise structure is also tailored for direct use with the variance-based bounds in our paper, as the term $|p(w|S) - p(w|S')|$ appears naturally in the Efron-Stein inequality analysis. In essence, our work proposes a stability definition that is not strictly "Bayesian" in the sense of Moran et al. (2023) but is a strong, distributional, and well-suited tool for our specific analytical pathway.

## G PROOF OF THEOREM 5.1

*Proof.* The theorem states that if a learning algorithm is $(\gamma_n, n)$-adaptively density stable, its generalization error for a $\sigma^2$-subgaussian loss is bounded by $\overline{\text{gen}} \leq \sqrt{2\sigma^2\gamma_n}$. The proof proceeds by first establishing a bound on the $V_2(S; W)$ measure under this stability condition, and then substituting this result into the intermediate bound from equation 47:

$$\overline{\text{gen}}\left(\mu, P_{W|S}\right) \leq \sqrt{\frac{\sigma^2}{n} V_2(S; W)}. \tag{60}$$

Our goal is therefore to show that Adaptive Density Stability implies $V_2(S; W) \leq \sqrt{2\gamma_n n}$.

$$\overline{\text{gen}} \leq \sqrt{\frac{\sigma^2}{n}} \, V_2(S; W), \tag{61}$$

Assume the definition of $V_2(S; W)$ as follows

$$V_2(S; W) = \mathbb{E}_{p_W}\left[\left(\mathbb{E}_{p_S}\left[\left(\frac{p_{W|S}(w|s)}{p_W(w)} - 1\right)^2\right]\right)^{1/2}\right]. \tag{62}$$

Then, applying the Efron-Stein inequality (see Boucheron et al., 2013, Chapter 3) yields

$$V_2(S; W) \leq \mathbb{E}_{p_W}\left[\left(\mathbb{E}_{p_S}\left[\sum_{i=1}^{n} \text{Var}_i\left(\frac{p(w|S)}{p(w)}\right)\right]\right)^{1/2}\right]. \tag{63}$$

If the stability condition

$$\frac{|p(w|S) - p(w|S')|}{p(w)} \leq \gamma_n, \tag{64}$$

holds for all neighboring datasets $S, S'$, then

$$\text{Var}_i\left(\frac{p(w|S)}{p(w)}\right) \leq \frac{\gamma_n^2}{4}. \tag{65}$$

This yields the intermediate bound

$$V_2(S; W) \leq \frac{\gamma_n \sqrt{n}}{2}. \tag{66}$$

To make our definition independent of the input distribution, which is unknown in practice, we further propose an extension using an arbitrary distribution $q(w)$

$$|p(w|S) - p(w|S')| \leq \gamma_n q(w) \quad \forall w, S, S'. \tag{67}$$

Building on ? (Theorem 33), we use their variational form for $V_2$

$$V_2(S; W) \leq \sqrt{\mathbb{E}_{p_S}\left[\sum_w \frac{p^2(w|S) - p^2(w)}{q(w)}\right]}. \tag{68}$$

Taking expectation over an independent copy $\tilde{S}$ of $S$ and using Jensen's inequality, we have

$$|p(w|S) - p(w)| = \left|p(w|S) - \mathbb{E}_{p_{\tilde{S}}}\left[p(w|\tilde{S})\right]\right| \leq \mathbb{E}_{p_{\tilde{S}}}\left[|p(w|S) - p(w|\tilde{S})|\right] \tag{69}$$

$$\leq \gamma_n \mathbb{E}_{p_{\tilde{S}}}\left[d_H(S, \tilde{S})\right] q(w) \leq \gamma_n n q(w). \tag{70}$$

where $d_H(S, \tilde{S})$ denotes the Hamming distance between datasets.

Combining these bounds gives

$$\mathbb{E}_{p_S}\left[\sum_w \frac{p^2(w|S) - p^2(w)}{q(w)}\right] \leq \mathbb{E}_{p_S}\left[2\gamma n \sum_w (p(w|S) + p(w))\right] \tag{71}$$

$$= 2\gamma_n n. \tag{72}$$

Thus we obtain the bound

$$V_2(S; W) \leq \sqrt{2\gamma_n n}. \tag{73}$$

Finally, substituting this into equation 61, we obtain

$$\overline{\text{gen}} \leq \sqrt{\frac{\sigma^2}{n}} \sqrt{2\gamma_n n} = \sqrt{2\sigma^2 \gamma_n}. \tag{74}$$

$\square$

# H    EXPERIMENTAL DETAILS AND ADDITIONAL RESULTS

This appendix provides a comprehensive description of the experimental setups and methodologies used in the paper, as well as additional simulation results on canonical channel models that provide further intuition for our theoretical bounds.

## H.1    DETAILS FOR THE BAYESIAN LINEAR REGRESSION EXPERIMENT

This section elaborates on the experiment presented in Section 6, which compares our proposed $V_2$ bound against several information-theoretic baselines.

### H.1.1 Simulation Setup

**Learning Task.** We consider a one-dimensional Bayesian linear regression problem. The data-generating process follows the linear model $y = w_{\text{true}}x + \epsilon$, where $w_{\text{true}}$ is the ground-truth weight, $x \sim \mathcal{N}(0,1)$ is the input feature, and $\epsilon \sim \mathcal{N}(0,\sigma^2)$ is i.i.d. Gaussian noise. To ensure the loss function is $\sigma_{\text{loss}}^2$-subgaussian, we employ a clipped squared error loss:

$$\ell(w,z) = \min\left((y-wx)^2, c\right), \tag{75}$$

where $z = (x,y)$ and $c > 0$ is a fixed clipping constant. This ensures the loss is bounded in the interval $[0, c]$. By Hoeffding's Lemma, the loss is therefore $\frac{c^2}{4}$-subgaussian.

**Learning Algorithm.** The algorithm is Bayesian linear regression. We place a zero-mean Gaussian prior on the weight parameter, $p(w) = \mathcal{N}(w|0, \lambda^{-1})$, where $\lambda$ is the prior precision. Given a training dataset $S = \{(x_i, y_i)\}_{i=1}^n$, the algorithm computes the posterior distribution $p(w|S)$, which, due to conjugacy, is also a Gaussian, $\mathcal{N}(w|m_n, s_n^2)$. The posterior parameters are given by the standard Bayesian update rules:

$$s_n^{-2} = \lambda + \beta \sum_{i=1}^n x_i^2 \quad \text{and} \quad m_n = s_n^2\left(\beta \sum_{i=1}^n x_i y_i\right), \tag{76}$$

where $\beta = 1/\sigma^2$ is the likelihood precision.

### H.1.2 Methodology for Estimating Bounds

All reported values for the empirical generalization error and the theoretical bounds are Monte Carlo estimates, averaged over $M$ independently generated training sets for each dataset size $n$.

**Empirical Generalization Error.** This serves as our ground truth. For each of the $M$ runs, a model was trained on a dataset $S_j$ to obtain the posterior $p(w|S_j)$. A single weight $w_j$ was drawn from this posterior. The generalization gap was calculated as the difference between the loss on a large, fixed test set (10,000 samples) and the loss on the training set $S_j$. The final reported value is the average of these gaps over all $M$ runs.

**The $V_2$ Bound (Theorem 4.1).** In the continuous setting, our proposed bound is $\overline{\text{gen}} \leq \int \sqrt{\frac{\sigma_{\text{loss}}^2}{n}\text{Var}_S[p(w|S)]}dw$. To estimate this, the integral was approximated by a sum over a fine grid of $K$ points, $\{w_k\}_{k=1}^K$. The core variance term, $\text{Var}_S[p(w_k|S)]$, was estimated for each grid point by computing the sample variance of the posterior probability density values, $\{p(w_k|S_1), \ldots, p(w_k|S_M)\}$, across the $M$ simulated training sets.

**MI Bound (Xu & Raginsky, 2017).** The mutual information bound is $|\mathbb{E}[\text{gen}]| \leq \sqrt{\frac{2\sigma_{\text{loss}}^2}{n}I(W;S)}$. The mutual information was calculated using the identity $I(W;S) = \mathbb{E}_S[\text{KL}(p(W|S)\|p(W))]$. For each of the $M$ runs, the KL divergence between the Gaussian posterior and the Gaussian prior was computed analytically. The final estimate is the average of these KL values.

**CMI Bound (Steinke & Zakynthinou, 2020).** We estimate the conditional mutual information bound using the supersample construction, where $I(W;U|\tilde{S}) = \mathbb{E}[\log p(W|S) - \log p(W|\tilde{S})]$. This was approximated using nested Monte Carlo. For each of $M_{\text{super}}$ outer-loop supersamples, we averaged over $J_{\text{inner}}$ inner-loop samples of training sets $S$ and hypotheses $W$. The marginal posterior $p(W|\tilde{S}) = \mathbb{E}_U[p(W|S_U)]$ was itself estimated with a Monte Carlo mixture of $K_{\text{marginal}}$ posteriors, computed stably using the log-sum-exp trick.

### H.1.3 Implementation Details

The simulation was implemented in Python 3. The specific parameters used in the experiment are listed in Table 1.

Table 1: Experimental Parameters for Bayesian Linear Regression

| Parameter | Description | Value |
|---|---|---|
| $w_{\text{true}}$ | True data-generating weight | 0.8 |
| $\sigma^2$ | Variance of the data noise $\epsilon$ | $0.4^2$ |
| $\lambda$ | Precision of the Gaussian prior on $w$ | 2.0 |
| $c$ | Clipping constant for the loss | 4.0 |
| $\sigma^2_{\text{loss}}$ | Sub-Gaussianity parameter of loss ($\frac{c^2}{4}$) | 4.0 |
| $n$ | Training dataset sizes | [10, 20, 40, 80, 160, 320] |
| $M$ | Number of Monte Carlo runs per $n$ | 1000 |
| $K$ | Number of grid points for $V_2$ bound | 500 |
| $[w_{\min}, w_{\max}]$ | Range of grid for $V_2$ bound | [-1.5, 1.5] |
| $M_{\text{super}}$ | Supersamples for CMI | 50 |
| $J_{\text{inner}}$ | Inner loops for CMI | 20 |
| $K_{\text{marginal}}$ | Samples for CMI marginal | 20 |

## H.2 ADDITIONAL RESULTS ON CANONICAL CHANNEL MODELS

To complement the main experiment, we provide additional simulations on three canonical channel models: the Binary Symmetric Channel (BSC), Binary Erasure Channel (BEC), and Z-Channel. These settings allow for exact analytical computation of the bounds. For these experiments, the dataset size is $n = 1$, the hypothesis space is $\mathcal{W} = \{0, 1\}$, and the loss is the squared error $\ell(w, z) = (w - z)^2$, which is bounded in $[0, 1]$ and is thus $\frac{1}{4}$-subgaussian. The term $\sqrt{\sigma^2/n}$ is omitted from the plots for clearer comparison.

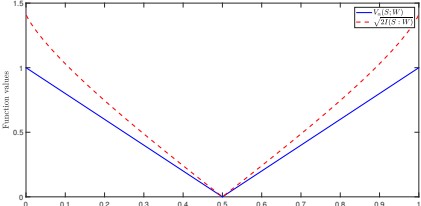

Figure 3: Comparison for the Binary Symmetric Channel (BSC). The $V_2$ bound is tighter than the MI bound across all crossover probabilities $p$.

Figure 4: Comparison for the Binary Erasure Channel (BEC). The $V_2$ bound is tighter than the MI bound across all erasure probabilities $\epsilon$.

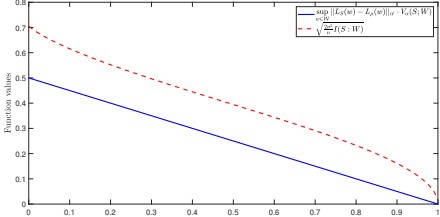

Figure 5: Comparison for the Z-Channel. The $V_2$ bound is again significantly tighter than the MI bound across all crossover probabilities $p$.

