# OpenReview forum: "One Measure, Many Bounds: Bridging TV, Variance, and Mutual Information"
_ICLR.cc/2026/Conference — Submitted to ICLR 2026_

### Official Review · Reviewer_fuvk · 2025-10-15

**Soundness:** 2
**Presentation:** 3
**Contribution:** 2
**Rating:** 2
**Confidence:** 5

**Summary:**

This paper introduces a new information-theoretic framework for generalization bounds based on an alternative for dependence measures $V_\alpha(S; W)$. It generalizes classical results built on total variation and mutual information, and at $\alpha = 2$ yields a novel variance-based bound that links the generalization gap directly to the variance of the algorithm’s output distribution. The authors further introduce Adaptive Density Stability as a sufficient condition for non-vacuous generalization and provide empirical validation on Bayesian linear regression, demonstrating that their bounds can be tighter than MI and CMI baselines.

**Strengths:**

* The introduced dependence measure $V_\alpha(S; W)$ bridges total variation, mutual information, and variance. It is both novel and unifying.
* The $\alpha=2$ case corresponds to a variance-based bound, which gives a direct, interpretable stability quantity and provides a data-dependent measure for generalization.
* The proposed ADS condition introduces a new way to ensure non-vacuous bounds, connecting distributional stability to practical learning rates.

**Weaknesses:**

Major:
* The related works discussed in the paper are a bit narrow and outdated. Only 2 baselines are considered, the MI bound from (Xu & Raginsky, 2017) and the CMI bound from (Steinke & Zakynthinou, 2020). There have been new bounds based on binary KL [1] or binary JS [2] comparators, and new individual information measures that provably lead to tighter bounds [3-5]. These works should be included in the comparison, and it is currently not clear if the proposed $V_\alpha$ surpasses these new methods or not.

Minor:
* The empirical evidence is limited to 1D Bayesian regression and simple channels. More real-world demonstrations would strengthen the paper’s impact.
* Typo: ? at L880.

[1] A new family of generalization bounds using samplewise evaluated CMI. 2022.

[2] Exactly Tight Information-theoretic Generalization Bounds via Binary Jensen-Shannon Divergence. 2025.

[3] Tightening mutual information-based bounds on generalization error. 2020.

[4] Sharpened generalization bounds based on conditional mutual information and an application to noisy, iterative algorithms. 2020.

[5] Individually conditional individual mutual information bound on generalization error. 2022.

**Questions:**

* Except for Figure 2, the comparison only involves $V_\alpha(S; W)$ and $I(W; S)$. I'll suggest including other related works (see weaknesses) in comparison, and report the standard deviation/confidence interval.
* I took a look at the source code, and am a bit confused about the choice of $\sigma$. The current implementation uses $\sigma = 1$ to compute the MI and CMI bounds, but uses a smaller $\sigma = 0.4$ to compute the $V_2$ bound, which may lead to unfair comparison. From my current reading (L950-960), they should be computed using the same $\sigma$. Can the authors justify this choice?

---

### Official Review · Reviewer_t6Vc · 2025-10-30

**Soundness:** 3
**Presentation:** 3
**Contribution:** 2
**Rating:** 4
**Confidence:** 4

**Summary:**

The work introduces a one-parameter family of information-theoretic generalization bounds based on the vector-valued Lp-norm correlation measure, Vα. While this framework unifies and interpolates between existing bounds such as those based on total variation and Rényi information. According to the authors, the primary conceptual contribution emerges at α = 2, where it yields a variance-based bound. This result establishes the variance of the algorithm’s output distribution, VarS[p(w|S)], as a direct, data-dependent measure of algorithmic stability, providing a new, information-theoretic perspective on how unstable (high-variance) algorithms fail to generalize.

**Strengths:**

- The paper is well-written and structured. It is easy to follow

- Understanding generalization is an important topic and deriving generalization bounds can provide new insights on when and why NN generalize.

- I checked the theoretical proofs of the paper and I think they are correct.

**Weaknesses:**

- **Limited Novelty/impact** My main concern with this paper is that the proposed bound does not provide any new insights. Although deriving a variance-based bound is interesting, the resulting formulation does not provide any new fundamental insights into generalization. In fact, it is well established in the literature that stability (and algorithmic stability) implies better generalization (see all works on algorithmic stabiity, e.g., Section 4.3 of [1]). While the authors redefine stability in terms of variance rather than sensitivity to the removal of one training sample (β-stability as in algorithmic stability), the two notions are equivalent as can be shown through the Efron–Stein inequality. Consequently, the proposed framework does not provide any new theoretical insights, it essentially reinterprets existing stability-based results in a different mathematical form (with a novel yet simple proof trick).

- **Incomplete coverage of related work:**  The authors fail to mention and discuss several related works that derive information-theoretic generalization bounds (e.g., [1–5]). In particular, [1] and [5] also provide stability-based analyses closely related to the one proposed here. [1-5] are only examples here. In fact, the literature in this area is now quite rich, and omitting prior works prevents readers from understanding how the current approach differs or improves upon prior efforts. I recommend that the authors expand the related work section to include and properly discuss all relevant recent studies.

- **Weak empirical validation:** The experimental results are confined to a Bayesian linear regression task and a few toy classification setups, which provide only preliminary evidence for the proposed framework. However, these tasks are simplistic and low-dimensional. There is no demonstration on real-world or deep learning scenarios, where the tightness or practical relevance of such bounds would matter most. Furthermore, comparing  only against two classical information-theoretic baselines (MI bound of Xu & Raginsky (2017) and CMI bound of Steinke & Zakynthinou (2020)) is insufficient. These are indeed foundational bounds, but the comparison omits more recent and stronger bounds, such as those from [1–5], which employ sample-wise or conditional decompositions and often achieve much tighter estimates in practice. Without benchmarking against these more contemporary methods, it is difficult to assess the true empirical or theoretical advantages of the proposed approach.




[1] Harutyunyan, Hrayr, et al. "Information-theoretic generalization bounds for black-box learning algorithms." Advances in Neural Information Processing Systems 34 (2021): 24670-24682. \
[2] Hellström, Fredrik, and Giuseppe Durisi. "A new family of generalization bounds using samplewise evaluated CMI." Advances in Neural Information Processing Systems 35 (2022): \
[3] Dong, Yuxin, et al. "Towards generalization beyond pointwise learning: A unified information-theoretic perspective." Forty-first International Conference on Machine Learning. 2024. \
[4] Wang, Ziqiao, and Yongyi Mao. "Generalization bounds via conditional $ f $-information." Advances in Neural Information Processing Systems 37 (2024): 52159-52188. \
[5] Wang, Ziqiao, and Yongyi Mao. "Sample-conditioned hypothesis stability sharpens information-theoretic generalization bounds." Advances in Neural Information Processing Systems 36 (2023): 49513-49541.

**Questions:**

See section above.

---

### Official Review · Reviewer_LQ2v · 2025-11-02

**Soundness:** 3
**Presentation:** 2
**Contribution:** 2
**Rating:** 2
**Confidence:** 4

**Summary:**

The paper introduces a new distribution- and algorithm-dependent complexity measure and uses it to derive several generalization bounds. The measure is intended to capture the dependence between the trained model W and the sample S:
V_alpha(S;W) = E_S [ ( E_{W|S} [ p_{W|S}(W|S) / p_W(W) ] - 1 )^(1/alpha) ].
For different alpha, the framework connects to TV- and MI-type bounds. The main contribution is a generalization bound based on V_alpha(S;W) (Theorem 3.1).

**Strengths:**

The strength of this work is that it proposes a new distribution and algorithm dependent complexity measure for generalization.

**Weaknesses:**

**MI scaling is not recovered.**
The paper suggests it “recovers” MI-style results, but the alpha = 1 specialization (Eq. 14) does not match the classical Xu–Raginsky rate:
|E[gen]| <= sqrt( (2*sigma^2 / n) * I(W;S) ).

As written, Eq. 14 has no explicit 1/sqrt(n) factor; since I(S;W) typically is constant, the RHS of Eq. 14 can only be constant and doesn't decrease with n.


**Eq. 15 is unclear or vacuous in rich or continuous hypothesis spaces.**
For continuous outputs, it’s not obvious how the sum/integral is defined or controlled; for large finite hypothesis classes, the term
sum_{w in W} sqrt( Var_S[ p_{W|S}(w|S) ] )
can scale poorly—contrast with the classical baseline sqrt( log|W| / n ).


**Computability/estimability of V_alpha**
All results hinge on access to p(W|S); outside simple models this is typically intractable.


**summary**
Without clear regimes where (i) the MI-type case exhibits 1/sqrt(n) behavior, (ii) Eq. 15 remains finite and scales with capacity, and (iii) V_alpha (or a surrogate) is computable in realistic pipelines, it’s hard to assess practical significance. Clarifying these points would substantially strengthen the paper.

**Questions:**

1- Please clarify in what sense  Eq. 14 is MI-type and state conditions under which the deviation term yields O(1/sqrt(n)).

2-Please specify conditions that make Eq. 15 finite. Also, how does it extends to continuous W.

---

### Official Review · Reviewer_26si · 2025-11-08

**Soundness:** 2
**Presentation:** 3
**Contribution:** 2
**Rating:** 4
**Confidence:** 4

**Summary:**

This paper introduces a novel, one-parameter family of information-theoretic generalization bounds based on the vector-valued $L_p$-norm correlation measure, $V_{\alpha}$. The key idea is that by tuning the parameter $\alpha$, this framework can interpolate between several existing types of generalization measures.

The paper highlights three key special cases:
1. $\alpha=1$: This regime recovers a well-known mutual information bound by Xu & Raginsky.
2. $\alpha \to \infty$: This regime leads to a worst-case deviation bound.
3. $\alpha=2$: This is presented as the main conceptual contribution. The framework yields a new, intuitive generalization bound that is directly controlled by the variance of the algorithm's output probability, $Var_S[p(w|S)]$.

Further, the authors introduce a new stability condition, "Adaptive Density Stability," to demonstrate a (strong) sufficient condition under which their bound is non-vacuous and achieves a standard $\mathcal{O}(1/\sqrt{n})$ learning rate.
Finally, the authors empirically validate their framework, demonstrating that their $V_2$ (variance-based) bound is tighter (on a single task) than a contemporary Conditional Mutual Information bound.

**Strengths:**

- The paper is well written.
- The primary strength of this paper is the introduction of an elegant framework ($V_{\alpha}$) that unifies multiple, previously distinct information-theoretic bounds
- The experiment in Section 6 (Figure 2) is compelling. Showing that the new $V_2$ bound is empirically tighter than the CMI bound from Steinke & Zakynthinou (2020) provides some evidence for the utility and non-triviality of this new measure. However, conclusively showing that this bound dominates the one by Steinke & Zakynthinou would require more experiments (even if they were simple examples).
- The $\alpha=2$ case, which directly links generalization error to the variance of the algorithm's output distribution ($Var_S[p(w|S)]$), is a significant and intuitive conceptual contribution.

**Weaknesses:**

- While Adaptive Density Stability is a useful theoretical tool, it presents a very strong, pointwise condition. The paper does not provide any concrete examples of common learning algorithms that are proven to satisfy this condition with the required $\gamma_n = \mathcal{O}(1/n)$ rate, which is necessary for the $\mathcal{O}(1/\sqrt{n})$ generalization bound.
- The framework's flexibility comes from the parameter $\alpha$. Figure 1 shows $\alpha=2$ as optimal for a simple Z-channel model. However, the paper does not provide a practical method or even a heuristic for selecting the optimal $\alpha$ for a given problem. Without this, it's unclear how to obtain the tightest possible bound from this framework in practice.
- The specialization at $\alpha=2$ (Theorem 4.1), which connects generalization to the algorithm's output variance $Var_{S}[p(w|S)]$, is a worthwhile contribution. However, the claim that this provides a fundamentally new information-theoretic perspective linking variance and stability to generalization may be slightly overstated (see questions).

**Questions:**

- See weaknesses.
- Your work presents a case for using the algorithm's output variance, $Var_{S}[p(w|S)]$, as a direct measure of stability that bounds the generalization error. This is a valuable contribution, particularly the novel variance-based bound derived at $\alpha=2$. I am curious how your stability measure (in the $\alpha=2$) relates to a different, recently proposed variance-based condition for learnability. This recent work (Proposition 5.2 in the 2024 paper by Gastpar et al.) has shown that an algorithm is “$l_2$-estimable” (meaning that there exists a tight bound (in terms of L2 error for its population risk) if and only if the conditional variance of its *population loss*, given the sample, is small (i.e., $\mathbb{E}[var(L_{\mathcal{D}}(A(S))|S)] \le \epsilon$). Your measure focuses on the variance of the algorithm's output probability over different samples $S$, while this other work focuses on the variance of the *outcome* (the true loss) over different distributions $\mathcal{D}$ that could have generated $S$. Could you comment on the relationship, if any, between these two distinct but related variance-based notions of stability? I believe that situating your work in relation to this alternative perspective on variance could further strengthen your paper's contribution.
- What is the missing reference in l. 881?



========================================================================

References:

Gastpar, M., Nachum, I., Shafer, J., & Weinberger, T.. Which Algorithms Have Tight Generalization Bounds?. Neurips 2025.

---

### Meta-Review · Area_Chair_cSrM · 2025-12-29

**Summary:**

This paper develops new information-theoretic generalization bounds based on a vector-valued $L_p$-norm correlation measure. While the reviewers acknowledge the novelty and potential generality of the proposed framework, they lean toward rejection mainly because the advantages of the resulting bounds are not fully convincing. In particular, the proposed bounds do not appear to recover the decay rate of the classical MI bound. Several reviewers also note that the bound or its key component seems difficult to evaluate beyond simple toy settings, which may limit practical applicability. In addition, there are concerns that the empirical comparisons focus only on older MI/CMI bounds, without benchmarking against more recent and stronger information-theoretic generalization bounds from the literature.

**Reviewer Concerns:**

No rebuttal was submitted, so the reviewers' concerns remain unaddressed.

**Reviewer Scores:**

Since the authors did not submit a rebuttal, it is very unlikely that any of the reviewers will change their score.

---

### Decision · Program_Chairs · 2026-01-26

Reject